



# The hourly wind-bias adjusted precipitation data set from the Environment and Climate Change Canada automated surface observation network (2001-2019)

Craig D. Smith[1], Eva Mekis[2], Megan Hartwell[2] and Amber Ross[1]

[1]Environment and Climate Change Canada, Climate Research Division, Saskatoon, SK
     [2]Environment and Climate Change Canada, Climate Research Division, Toronto, ON

*Correspondence to:* Craig D. Smith (craig.smith@ec.gc.ca)

**Abstract.** The measurement of precipitation in the Environment and Climate Change Canada (ECCC) surface network is a crucial component for climate and weather monitoring, flood and water resource forecasting, numerical weather prediction and many other applications that impact the health and safety of Canadians. Through the late 1990s and
early 2000s, ECCC surface network modernization resulted in a shift from manual to automated precipitation measurements. Although many advantages to automation are realized, such as enhanced capabilities for monitoring in remote locations and higher frequency of observations at lower cost, the increased reliance on automated precipitation gauges has also resulted in additional challenges, especially with data quality and homogenization. The automated weighing precipitation gauges used in the ECCC operational network have an increased propensity for
wind-induced undercatch of solid precipitation. One outcome of the WMO Solid Precipitation Inter-Comparison Experiment (SPICE) was the development of transfer functions for the adjustment of high frequency solid precipitation measurements made with gauge/wind shield configurations used in the ECCC surface network. Using the SPICE Universal Transfer Function (UTF), hourly precipitation measurements from 397 ECCC automated climate stations were retroactively adjusted for wind undercatch. The data format, quality control and adjustment procedures are
described here. The hourly adjusted data set (2001-2019, version v2019UTF) is available via the ECCC data catalogue: https://doi.org/10.18164/6b90d130-4e73-422a-9374-07a2437d7e52 (ECCC, 2021). A basic spatial impact assessment shows that the highest relative total precipitation adjustments occur in the Arctic where solid precipitation has an overall higher annual occurrence ratio. The highest adjustments for solid precipitation are shared by the Arctic, southern Prairies and the coastal Maritimes, where stations tend to be more exposed and snowfall events occur at
higher wind speeds.

## 1 Introduction

### 1.1 Motivation

Accurate precipitation measurements are required for many climatological, meteorological and hydrological applications such as climate trend analysis (Vincent and Mekis, 2006; Vincent et.al, 2018), water resource forecasting

(Barnett et al., 2005; Pomeroy et al., 2007), and numerical weather prediction model verification (Buisán et al, 2020; Køltzow et al, 2020). It is well recognized that precipitation observations made in cold regions are prone to underestimation due to the undercatch of solid precipitation related to gauge configuration and wind speed (e.g. Sevruk et al., 1991; Goodison et al., 1998; Kochendorfer et al., 2017a).

The Environment and Climate Change Canada (ECCC) surface observation network has undergone many historical transitions but perhaps the most significant is the shift from manual to automated measurements. This transition began in the early 1990's and accelerated in the early 2000s (Fig. 1, Mekis et al., 2018). The transition to automated precipitation measurements has both advantages and disadvantages. Although automation of stations is a cost-saving

10 alternative, the current state of technology is such that this practice imposes limitations in terms of data quality and availability (such as the lack of independent snowfall observations). Furthermore, the lack of detailed metadata limits the possibilities for correcting some of the observations (Mekis et al, 2018). Automation does provide the potential for lower cost monitoring in remote regions with increased frequency of observations (i.e. hourly rather than daily measurements).

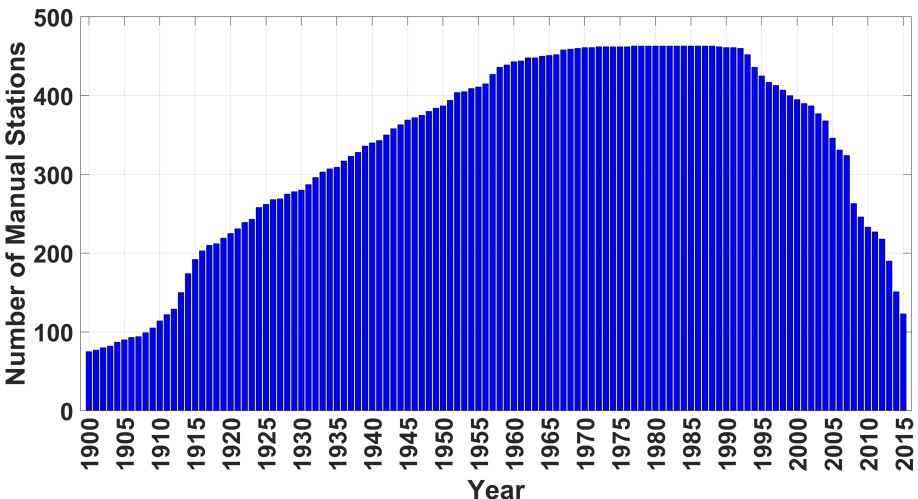

**Figure 1: Evolution of the quantity of manual stations in the ECCC operational network from 1900 to 2015 (from Mekis et al., 2018).**

Along with station automation came the change in instrument configuration for the measurement of precipitation. While the use of the manual Type B gauges for measuring liquid precipitation and the snow ruler or Canadian Nipher gauge for measuring solid and mixed precipitation were prevalent into the mid-1990s (Devine and Mekis, 2008), the

use of automated gauges such as the Belfort Universal and Fisher and Porter (which appeared as early as 1965) became more common. Eventually, the Geonor T-200B, OTT Pluvio[1] and the OTT Pluvio[2]L became the standard gauge configuration in the ECCC automated network (Mekis et al., 2018). Gauge intercomparisons as early as the 1970's and 1980's (Goodison, 1978; Goodison et al., 1998) showed that catch efficiency of these automated configurations

(both shielded and unshielded) for solid precipitation decreased with wind speed much faster than for the Canadian Nipher gauge. The resulting inhomogeneities introduced into the Canadian observation network with automated gauges are problematic for climate trend analysis, but historically, the impact of these inhomogeneities were mitigated by employing climate dependent conversions to manual snow ruler measurements as a proxy for the gauge measurement of solid precipitation (Mekis and Brown, 2010; Mekis and Vincent, 2011). However, this technique is

now threatened by the continued decline in manual snow ruler snowfall measurements in much of Canada, which hastens the need to utilize the automated precipitation gauge measurements from the in situ ECCC observation network. In turn, this necessitates the adjustment of those measurements for the systematic bias due to wind induced undercatch of solid precipitation.

**1.2 WMO-SPICE recommendations**

One of the key objectives of the World Meteorological Organization (WMO) Solid Precipitation Inter-Comparison Experiment (SPICE) was to:

assess the possibility of deriving transfer functions to account for (and, ideally, to correct) wind-induced error in solid precipitation measurement and this included investigating the concept of "universal" transfer functions, which can be applied to data from instruments with a specific configuration in different climate conditions (Nitu et al., 2018).

Following SPICE analysis, and the development of a "universal" transfer function (hereafter referred to as UTF), the

recommendation from SPICE is to apply the methods for developing transfer functions and apply those transfer functions to national and regional precipitation measurements in various climate regimes, while recognizing that more work is necessary for the development of climate zone specific transfer functions (Nitu et al., 2018).

**1.3 The SPICE Universal Transfer Function**

The development of the SPICE UTF for adjusting the automated accumulating gauge measurements of solid precipitation followed the development of previous transfer functions that were generally more focused on site specific gauge intercomparisons (Smith, 2008; Wolff et al., 2015; Kochendorfer et al., 2017a). What makes the SPICE UTF unique is the methodology that combines the catch efficiency ratios (configuration under test vs. the reference) for automated gauges using the same shield configuration at multiple (eight) sites. Kochendorfer et al. (2017b) describes

this methodology, and the resulting transfer functions, in more detail. The methodology employed the WMO Double



Fence Automated Reference (DFAR) system, which incorporated a precipitation detection device to increase the reliability of the 30-min event based data used to fit the transfer function model. There were two transfer function models published by Kochendorfer et al. (2017b), one that is continuous with temperature and the other requiring the user to determine precipitation phase. Since there are no direct observations of phase available operationally in Canada

via the automated networks (and determining phase would require user subjectivity), the temperature continuous transfer function model was chosen for this application. This transfer function takes the form:

$$C_E = e^{-a(U)(1-\tan^{-1}(b(T_{air}))+c)} \tag{1}$$

where $C_E$ is the catch efficiency of the automated gauge, U is wind speed (either measured at the standard 10 m height or at gauge height) in m s$^{-1}$, $T_{air}$ is air temperature in degrees C, and $a$, $b$, and $c$ are coefficients to fit the data to the model. In the SPICE methodology, to reduce the impact of fewer events at higher wind speeds and the potential impacts of blowing snow, the SPICE data were filtered to remove 30-min precipitation events with mean wind speeds higher than a threshold ($U_t$) of 7.2 m s$^{-1}$ (9 m s$^{-1}$) at gauge height (10 m). The transfer function for a single Alter-

shielded gauge is shown in Fig. 2 using the coefficients from Kochendorfer et al. (2017b) (Table 1) for wind speed measured at gauge height (typically 2 m above ground level) and at selected distinct temperatures between +5 °C and -20 °C.

**Table 1: Coefficients *a*, *b*, and *c* used in Eq. 1 specified for a single Alter-shielded gauge with wind speed measured at gauge**
**height ($U_{gh}$) and at 10 m ($U_{10}$)  (from Kochendorfer et al., 2017b).**

| Wind Speed Height | a | b | c | $U_t$ |
|---|---|---|---|---|
| $U_{gh}$ | 0.0348 | 1.366 | 0.779 | 7.2 m s$^{-1}$ |
| $U_{10}$ | 0.0281 | 1.628 | 0.837 | 9.0 m s$^{-1}$ |

A performance assessment was completed on the UTF by both Kochendorfer et al. (2017b) and Smith et al. (2020). Kochendorfer et al. (2017b), using data from the same intercomparison period as used for the UTF development, showed that application of the UTF to individual sites reduced the bias in the precipitation measurement but did not
necessarily decrease the uncertainty. Similar to Kochendorfer et al. (2017b), Smith et al. (2020) used a more independent (post-SPICE) assessment data set and showed that the performance of the UTF varied substantially with site, generally under-adjusting solid precipitation measurements at windy sites and over-adjusting solid precipitation measurements at less windy sites (using a 10 m mean wind speed during snowfall of approximately 3 m s$^{-1}$ to divide the sites into the two wind classes). The message from both assessments is that the adjustments should be used with
some caution, recognizing that site climatic conditions impact the UTF performance and that this impact is difficult to assess without a co-located reference.

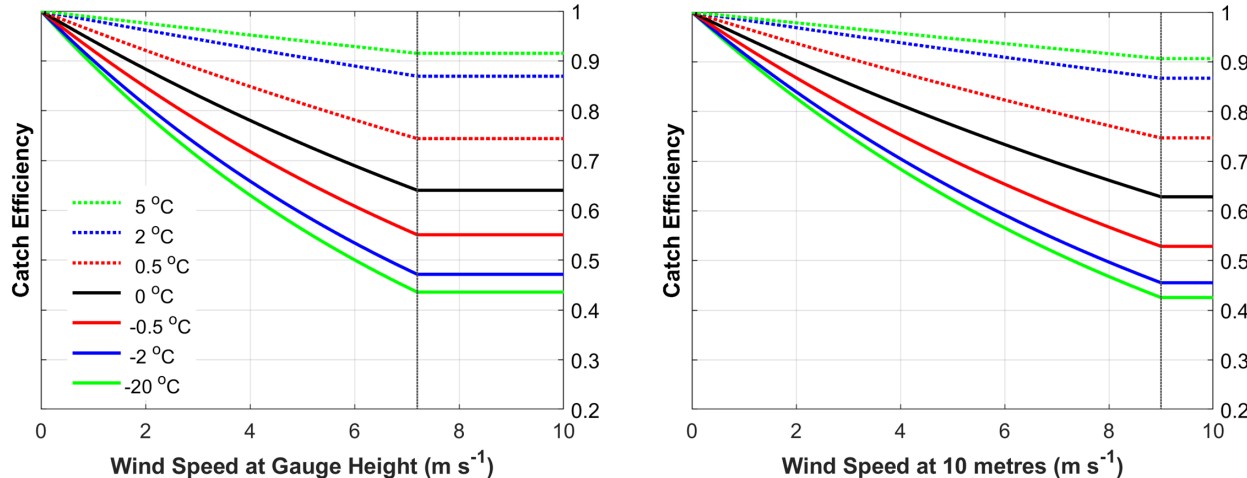

**Figure 2: The SPICE Universal Transfer Function (Eq. 1) plotted with coefficients for the single Alter-shield and wind measured at gauge height (left) and 10 m (right) for temperatures of +5, +2, +0.5, 0, -0.5, -2, and -20 °C. The inflection point where catch efficiency is set to a minimum occurs at wind speed thresholds ($U_t$) of 7.2 m s$^{-1}$ (for gauge height wind) and 9.0 m s$^{-1}$ (for 10 m wind) and is marked by the vertical dotted line.**

## 2 Data description and methods

The stations list for the hourly wind-bias adjusted precipitation data set is a subset of the stations in the automatic

5   station network operated by ECCC. The network consists of fully automated stations, including both Surface Weather and Reference Climate Stations (RCS). Special focus was given to the RCS "protected" stations, i.e. those identified by ECCC to the WMO for longevity and reliability (Vincent, 2020) to fulfill climate research focused requirements in data-sparse areas of Canada. The typical observed parameters are air temperature, precipitation accumulation and intensity, wind speed and direction, snow depth, humidity and air pressure. Most locations are inspected by ECCC on

10  a regular (annual or semi-annual) basis.

All hourly precipitation, temperature and wind observations were extracted from the Historical Climate Data repository of ECCC's National Climate Archive (http://www.climate.weather.gc.ca).

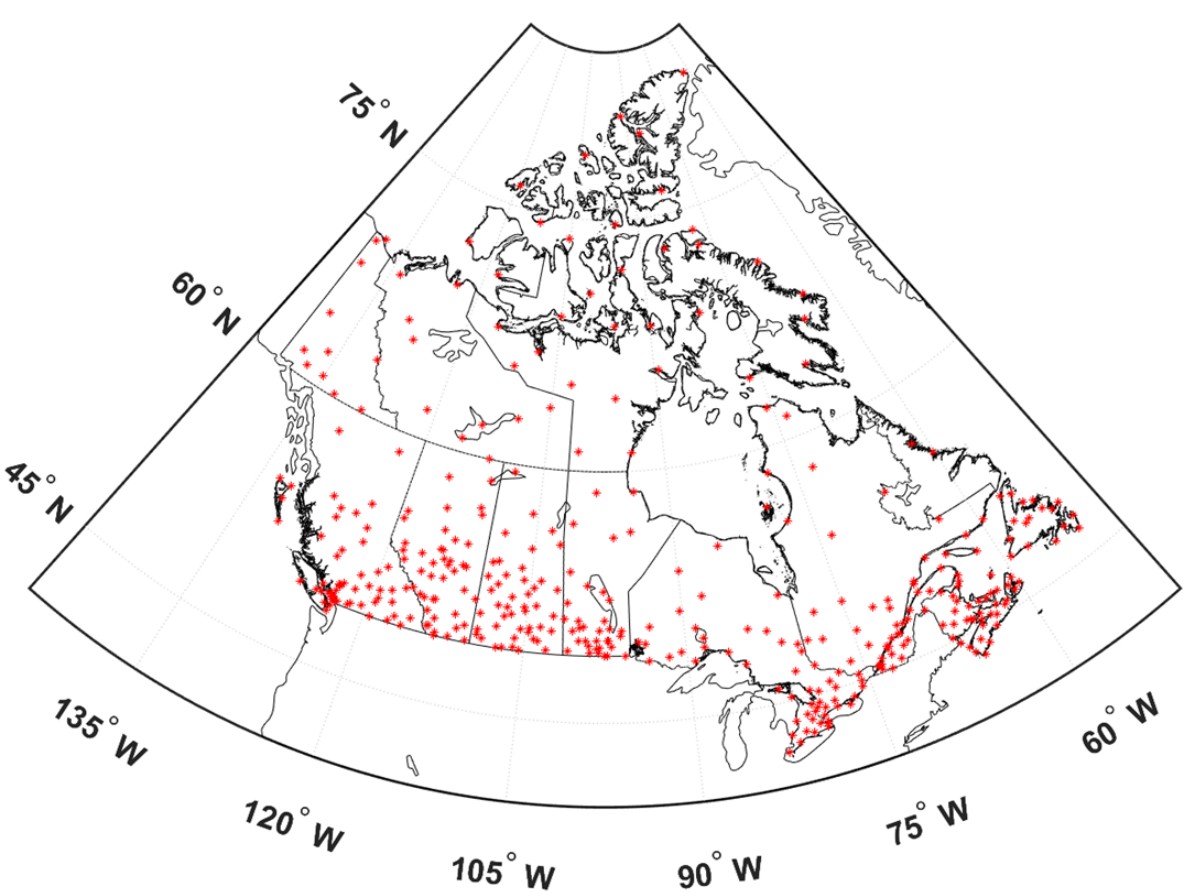

**Figure 3: Location of the 397 ECCC automated climate stations selected for the application of the SPICE UTF.**

### 2.1 Site selection

Initial station selection included all Geonor and Pluvio gauges that were installed prior to 2015 (Milewska et al., 2019).
This included many stations in northern Canada, where automation started in the early 2000's. A later set of stations
was added to this list, which were selected for their value to climate research. These stations were selected based on
their proximity to adjusted and homogenized Canadian climate data (AHCCD, Mekis and Vincent, 2011) stations, as
well as data quality and site longevity. An effort was made to include all RCS locations, as they are designated as
"protected" sites.

Earth System
Science
Data

## 2.2 Precipitation measurements

The first version of this adjusted automated data set, published by Milewska et al. (2019) used a transfer function developed prior to WMO-SPICE and included 312 Geonor T-200B (600 mm capacity) and 34 OTT Pluvio[1] (1000 mm capacity) or Pluvio[2]L (1500 mm capacity) automated gauges. However, the gauge types deployed in the network

5     change over time, with stations being updated with new instruments from different manufacturers. The most significant network modernization effort is the gradual replacement of the single Alter-shielded Geonor T-200B (Fig. 4 left) with the double Alter-shielded OTT Pluvio[2]L (Fig. 4 right)  gauge, resulting in the evolution of the ratio of Geonor to Pluvio[2]L gauges in the network. This change has implications on catch efficiency varying with location based on double Alter-shield deployment and impacting adjustment procedures (discussed in later sections).

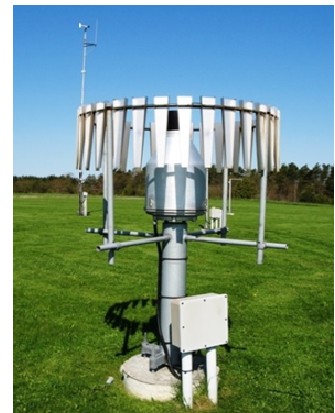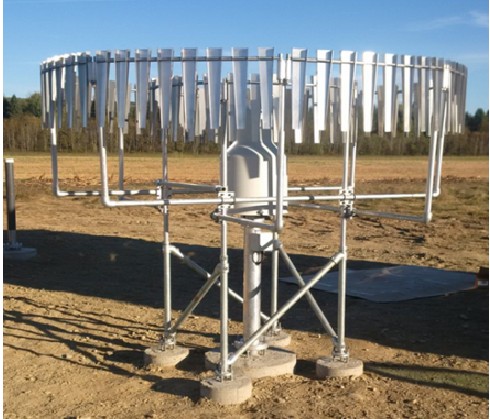

**Figure 4: Standard automated precipitation gauge configurations deployed across the ECCC network: Geonor T-200B gauge with Single Alter-shield (left) and OTT Pluvio[2]L gauge with double Alter-shield (right).**

The Geonor T-200B weighing gauge operates using a collecting bucket suspended in a tray connected to the frame by three vibrating wire transducers. The bucket weight (and equivalent precipitation accumulation) is calculated by applying a quadratic calibration function to the measured vibration frequency for each transducer, sampled every 5

15     seconds over the last 5 minutes of every 15-min period. The data logger records raw bucket weight, filtered bucket weight, and total precipitation calculated as the filtered bucket weight differential from the previous 15-min period, for each of the three vibrating wire transducers.

The similarly shaped OTT Pluvio[1] and Pluvio[2]L precipitation gauges operate by measuring the bucket weight with a load cell. The older generation Pluvio[1] uses an internal processor to produce a pulse output for each 0.1 mm

20     equivalency increase in the bucket weight, which is then interpreted as a precipitation accumulation. The sensor's 0.1





mm pulses are sampled once every 5 seconds and the number of pulses are summed to calculate the 15- and 60-min total precipitation. The Pluvio²L is a newer generation of this instrument with more complex on-board algorithms and the capability of digital output. The Pluvio²L digital output of accumulated precipitation amount is sampled and logged every 15 minutes and interval precipitation calculated as the difference between the current and previous amount.

Digital data processing is applied to all automated gauges. Further processing details, including processing and output characteristics, are summarized in Mekis et al. (2018). The generated output fields are controlled and stored by on-site data loggers (Campbell Scientific CR3000 or legacy CR23X) via nationally deployed data logger software programs which also perform basic sampling and filtering tasks at each location. Depending on the gauge configuration, a logger-based filter is used to minimize signal noise and false precipitation reports and is set at a

threshold based on the minimum measurable amount for each gauge type (0.2 mm for the Geonor and 0.1 mm for the Pluvio²L).

### 2.3 Wind speed and temperature measurements

For the adjustment procedure (see Sect. 1.3 and Eq. 1), good quality and reliable hourly temperature and wind observations are required. Similar to the precipitation, the hourly surface temperature and wind observations across

Canada were retrieved from the Historical Climate Data repository of ECCC's National Climate Archive (https://climate.weather.gc.ca/index_e.html).

From the National Archive, the following information were used:

- Element number 078: hourly surface dry bulb temperatures (0.1 °C) measured inside MSC Stevenson Screens by the YSI 44212EC temperature sensors.

- Element number 076: hourly wind speed (0.1 km h⁻¹) measured with a Campbell Scientific wind monitor (mainly model # 05103, https://www.campbellsci.ca/05103-10 but can have local variations) at 10 m heights. In the aviation network, this observation is a 2-min mean averaged prior to the top of the hour. At the MSC automatic stations, this is a 10-min mean, averaged prior to the top of the hour. Neither observations represent the mean wind speed for the full hour.

- Element numbers 271-274: these are 15-min wind speed (0.1 km h⁻¹) averages measured at gauge height (approximately 2 m above ground) for minutes 00-15, 15-30, 30-45 and 45-60 minutes respectively. This instrument is also mainly a Campbell Scientific model #05103 wind monitor. Unlike element number 076, elements 271-274 represent the mean for the period which they represent. The four 15-min elements are averaged to produce an hourly mean which is used in the adjustment.



Both wind speed measurements (element 076 at 10 m and elements 271-274 at gauge height) are required for this adjustment exercise. The gauge height wind is the preferred measurement for use in Eq. 1, but it is unfortunately not always available. The selection process and data flagging for wind are explained further in Sect. 2.5 and 2.6.

**2.4 Data quality control and CODECON correction**

After the initial filtering by the data logger, the precipitation data is processed through the ECCC Data Management System (DMS) and then stored in the National Climate Data Archives. The required hourly and 15-min values are extracted directly from the archive, and subjected to enhanced quality control (QC) procedures.

The first enhanced QC procedure for precipitation is a range check against a fixed threshold to screen extraneously large values. This threshold is set to 75 mm for 15-minute values and 110 mm for hourly values, based on the Canadian

all-time records in these respective observation periods (Milewska et al. 2019). Next, the data is checked for negative values, which could occur when the gauge is being serviced or through filtering errors at the data logger. In this particular precipitation data set, many small negative values of -0.1 mm needed to be removed, occurring primarily from the Geonor gauges prior to 2014. Similar to this, the entire data are filtered for precipitation observations that are less than a pre-determined noise threshold for a particular gauge. These small amounts are too low to be

distinguished from noise and the thresholds correspond to the reporting limits for each gauge (Nitu et al., 2018). This also resulted in the removal of many 0.1 mm values from the Geonor gauges prior to 2014 due to the gauge reporting threshold of 0.2 mm. No small values needed to be removed from the OTT Pluvio (generation 1 and 2) gauges due to their higher precision reporting threshold of 0.1 mm.

Following the automated QC, the data are then screened manually, under the guidance of ECCC regional

climatologists that provide local context. The data were first examined visually to detect large values when compared to the daily total precipitation, as well as the monthly ratios of the 15-minute and hourly totals compared to the daily totals. These checks were applied to data collected in 2013 and earlier. Subsequent checks of the hourly values were performed, in comparison to the spread around the station mean. An hourly value was flagged and manually inspected if the value exceeded 11 standard deviations (SD) from the station's mean value. The decision on the verity of the

hourly value was made based on comparing it to the daily value, neighbouring stations, historical radar and station IDF curves; over 1000 suspicious values were checked in this manor and approximately 68% of these values were removed.

A final manual check of the monthly total precipitation values was performed using the de-seasonalized time series. High values were identified as exceeding 4 SD from the monthly de-seasonalized mean. All flagged months were

inspected at the hourly level and any erroneous hourly data were removed.

Some basic quality control was performed on the hourly temperature and wind speed values after they were extracted from the archive. The hourly temperatures were identified as possible outliers if they were below –55 °C or above 40

°C. None of these values were deemed erroneous, and all were retained. The hourly wind speed was replaced with a missing value if it was below 0 km h$^{-1}$ or above 150 km h$^{-1}$; there were only a handful of occurrences of the high wind speed. If 12 or more consecutive hours reported a wind speed of exactly 0 km h$^{-1}$, then the full string is replaced as missing; this is likely caused by a seized anemometer (due to frost or other mechanisms).

Another revision to the archived precipitation data was required to fix issues created by a legacy data filter designed to remove relatively small noise amounts recorded by older generation automatic precipitation gauges that were prone to issues such as wind pumping (where airflow over the gauge causes the spring weighing mechanism to bounce and register false precipitation amounts). This filter was part of a data management system called Code Conversion (CODECON) that predated the current DMS and remained in place for several years following the replacement of the

older generation precipitation gauges with models that were less prone to wind pumping errors. The result was the erroneous filtering of small but realistic precipitation amounts observed by the newer gauges capable of observations at much higher resolutions (i.e. tenths rather than whole mm).  Henceforth, the filter will be called the CODECON filter and the error adjustment called the CODECON correction, both are explained in greater detail by Milewska et al. (2019).

The CODECON filter was in place up until December 2013 and impacted the hourly accumulated total precipitation amounts saved in the archive. The 15-min precipitation values were not affected by the CODECON filter, so when available (availability being temporally and spatially variable), these were used to compute the hourly precipitation values, effectively bypassing the CODECON filtering issue. However, when and where the 15-min data was unavailable, an alternative correction methodology was required. Firstly, stations with overlapping 15-min and hourly

precipitation values were used to quantify the impact of the CODECON filter on the scale of monthly precipitation, and this amount was regionally averaged to be used as a correction for each hour with precipitation. This amount was calculated for each region and month, and then divided between all hours with > 0 mm precipitation at stations without 15-min observations.

### 2.5  Applying the Universal Transfer Function

The adjustment to the hourly precipitation measurements are completed using Eq. 1 and the coefficients shown in Table 1 (plotted in Fig. 2), depending on the available wind speed measurement.  The default is to use the gauge height hourly mean (average of elements 271-274) wind speed measurement when available, but the adjustment reverts to the 10 m (element 076) wind speed measurement if the gauge height measurement is not available. The availability information for the wind speed measurement used in the adjustment is declared via a flag for each hourly observation.

If either the air temperature or the wind speed measurement is missing, no adjustment is made to the precipitation measurement. However, if either ancillary measurement is missing but there is a valid unadjusted precipitation measurement, the unadjusted measurement is presented and flagged accordingly.

Since Eq. 1 is continuous with temperature, an adjustment to a precipitation event above 0 °C can occur. This allows for the adjustment of solid and mixed precipitation that occurs when near surface air temperatures are above freezing. However, adjustments can conceivably occur at higher temperatures when precipitation phase is almost definitely rain. Although Eq. 1 approaches 1 at higher temperatures, there is a small potential for over-adjustment of rain at high

wind speeds. For this reason, $C_E$ is set to 1 at all temperatures greater than 5 °C.

The estimated CODECON correction explained above, if required, is added to the precipitation measurement following the adjustment. This sequence of adjustment was chosen to avoid compounding uncertainty related to both the application of the transfer function and the estimate of the CODECON correction. The data flags related to the transfer function adjustment are described in the next section and exceptions for adjustments related to the ancillary

data or the gauge configuration are noted in the discussion (Sect. 3).

### 2.6  Data format and flagging

#### 2.6.1  File naming and format

The data files are ASCII space delimited text files with one file for each of the 397 stations. The file naming nomenclature is as follows: #######_UTF_hly_prec.txt, where ######## is the 7-digit station identifier. A comma

delimited station catalogue contains the station identifiers, station names, province of installation, spatial coordinates, and the start and end dates of the station data. This catalogue file is stored in the linked data archive.

#### 2.6.2  Data description

Data files contain hourly data and consist of 2 header rows (one in English and one in French declaring the variable name and the units) and 10 columns: 1 timestamp, 2 flag, and 7 data columns. The timestamp is in local standard time

(consistent with the ECCC archive) using ISO 8601 formatting: YYYYMMDDThhmm (where "YYYY" the year, "MM" the month, "DD" the day, and the time of day follows "T" as the hour "hh" and the minute "mm"). The timestamp represents the end of the accumulation or averaging period. Table 2 describes the content of the 10 columns, and the following section describes the flagging.





**Table 2: Data file column descriptions**

| Column | Header (English) (French) | Description |
|---|---|---|
| 1 | YYYYMMDDThhmm<br>YYYYMMDDThhmm | Time stamp in LST, YYYY=year, MM=month, DD=day, Thhmm=time in hours(hh) and minutes(mm). |
| 2 | Unadj_P(mm)<br>Nonaj_P(mm) | Unadjusted precipitation in millimetres that occurred over the previous 60 minutes. |
| 3 | Tair(C)<br>Tair(C) | Air temperature in degrees Celsius measured at the top of the hour. |
| 4 | Wind(m/s)<br>Vent(m/s) | Wind speed in metres per second as measured at 10 m or gauge height as indicated by the flag in Column 5. |
| 5 | Wind_Flag<br>Vent_Ind | Flag declaring the measurement height of the wind speed variable. Column 5 flags are defined in Table 3. |
| 6 | CE<br>EC | Catch efficiency ($C_E$) expressed as a ratio, calculated by Eq. 1. |
| 7 | UTF_Adj_P(mm)<br>UTF_Aju_P(mm) | The adjusted precipitation amount in millimetres derived as Column 2 x Column $6^{-1}$ or as Column 2 when there is no adjustment. |
| 8 | CODECON(mm)<br>CODECON(mm) | The CODECON correction in millimetres to be added to the final precipitation amount (if required). |
| 9 | UTF_Adj+CODECON_P(mm)<br>UTF_Aju+CODECON_P(mm) | Final precipitation amount in millimetres, derived from adding Column 7 and Column 8. |
| 10 | Adj_Flag<br>Aju_Ind | Flag describing the adjustment. Column 10 flags are defined in Table 4. |



### 2.6.3 Flagging

Two flagging columns appear in the data: columns 5 and 10. Column 5 flags the wind speed measurement height used in Eq. 1 to calculate the catch efficiency, and they are defined in Table 3. The flag in column 10 describes the adjustment. These are defined in Table 4. All flags are discussed further in Sect. 3.

5 **Table 3: Wind speed measurement flags (Column 5)**

| Flag | Description |
|------|-------------|
| 2 | Wind speed is measured at gauge height, approximately 2 m above the ground |
| 10 | Wind speed is measured at 10 m above the ground |
| 8 | Wind Speed is missing (-99999) |

**Table 4: Precipitation adjustment flags (Column 10)**

| Flag | Description |
|------|-------------|
| 1 | $C_E$ from Eq. 1 is greater than or equal to 1 and no adjustment is performed |
| 2 | $C_E$ from Eq. 1 is less than 1 and data is adjusted using this value |
| 3 | Not used |
| 4 | Precipitation is adjusted but should be used with caution since wind speed is measured at a neighbouring site at a distance less than 12 km |
| 5 | Not used |
| 6 | Precipitation is measured with a gauge using a double Alter-shield; precipitation is not adjusted |
| 7 | Precipitation is from a gauge/shield that can not or should not be adjusted (other than a double Alter-shielded gauge) |
| 8 | Temperature or wind data are missing (-99999); precipitation is not adjusted |
| 9 | Precipitation is missing (-99999); not adjusted |

### 3 Discussion

10 ### 3.1 Adjustments and exceptions

Extensive effort has been made to produce a complete adjusted precipitation data set but there are limitations to the adjustment capabilities of the generalized SPICE UTF. The UTF equation (Eq. 1) was developed for single Alter-shielded automated precipitation gauges, predominately Geonor T-200B or OTT Pluvio[2] gauges because these gauges were more widely deployed during the SPICE project. Hence, we have the most confidence in applying the SPICE

15 UTF to similar automated gauge configurations. For the most part, the single Alter-shielded Geonor and Pluvio[2]L comprise the majority of gauges in this data set. The exceptions, with the approximate number of each deployed in



the network shown in parentheses (as of 2019), include single Alter-shielded Belfort Universal (10), Tretyakov or unshielded Pluvio[1] (31), Geonor high capacity T-200B (2), and double Alter-shielded Pluvio[2]L (68) gauges. The high capacity single Alter-shielded Geonor T-200B and Belfort Universal gauges, because of the applied common wind shield, are expected to have similar $C_E$ to the single Alter-shielded gauges assessed during SPICE. This expectation

can be justified via previous studies showing that in determining $C_E$ the shield configuration is more important than the actual gauge inside the shield (Yang et al., 1999; Rasmussen et al., 2012; Kochendorfer et al., 2017b). Therefore all measurements from the single Alter-shielded Geonor T-200B and Belfort Universal gauges are adjusted using the UTF. The Tretyakov and unshielded Pluvio[1] gauges are mostly deployed in the province of British Columbia (with 1 in each of the Yukon and Northwest Territories and 2 in Ontario). Since there is not a TF for these automated

configurations, they have not been adjusted and the data from these gauges are flagged with a number "7" (Table 4).

There are 4 stations measuring precipitation without co-located wind speed data available for a portion of the record, where a neighboring site (< 12 km distant) is used. These precipitation measurements are adjusted but flagged with a number "4" indicating that they should be used with caution. This occurs at Kelowna UBCO (ID 1123996), Vancouver Harbour (ID 1108446), Pemberton Airport (ID 1086082), and Edmonton International Airport (ID 3012205).

Measurements made with the double Alter-shielded Pluvio[2]L are also not adjusted in this data set. The cumulative number of effected stations, extracted from the network metadata (Jan 2021), are displayed in Fig. 5, although the metadata records for the network (dates and numbers of double Alter-shield installations) are somewhat dynamic due to an ongoing metadata database revision exercise. It is well understood that double shielding can substantially increase the catch of solid precipitation (Watson et al., 2008; Kochendorfer et al., 2018, Nitu et al., 2018) as compared

to a single Alter-shielded gauge configuration. The assumption was made that using the SPICE UTF (Eq. 1) with the single-Alter coefficients (Table 1) to adjust data collected with a double Alter-shielded gauge would result in a substantial over-adjustment. The date of the double Alter-shield installation was derived from the station metadata and the Pluvio[2]L measurements were not adjusted after this date. This unadjusted data is flagged with a number "6". Although Kochendorfer et al. (2018) did present a TF for adjusting double Alter-shielded gauges, this TF is developed

with very limited data and assessment, and is therefore not used here. As the number of double Alter-shielded gauges increase in the network (post-2019), an appropriate TF will need to be developed and applied.

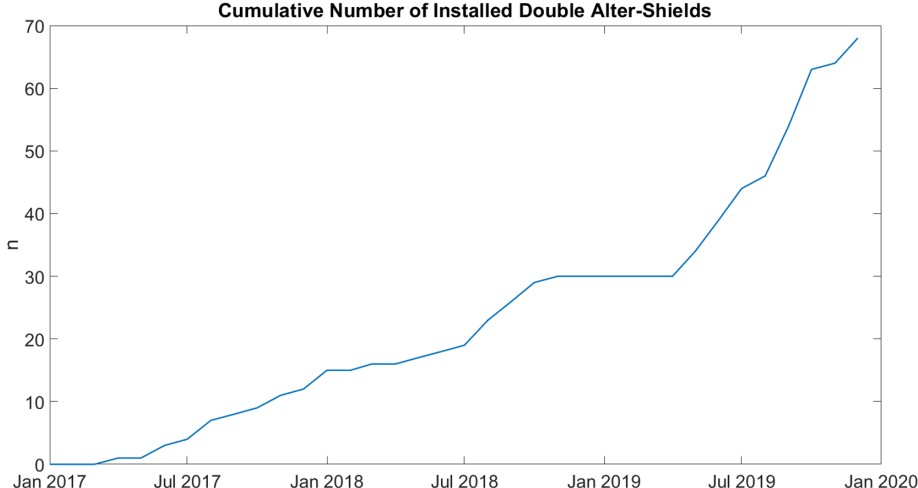

**Figure 5: Cumulative total number of Pluvio²L double Alter-shielded gauges installed in this subset (n=397) of stations in the ECCC operational network from January 2017 through December 2019.**

### 3.2 Impacts of adjustments on regional precipitation totals

The impact of the wind-bias adjustment on measured precipitation, by station, falling as solid (snow) and as total precipitation are shown in Fig. 6. The regional impact on the measurement of precipitation falling as snow (Fig. 6 top; hourly mean $T_{air} < -2$ °C) is apparent in the Arctic, southern Prairies and along the east coast of the Maritimes. These are generally stations with high exposure and high wind speeds during snowfall events. While solid precipitation after adjustment increases by nearly 75% at some Prairie stations, the change is more than double at several Maritime stations. The impact of adjustments on total precipitation are more muted, especially at stations where the solid to total precipitation ratio is low (west coast, south-central, and southern Maritime regions). The largest impacts are shown in the Arctic where snowfall is a larger portion of total precipitation, with Arctic stations showing a general increase between 20% and 40% while some Arctic stations show over 50% total increase.

### 3.3 Uncertainty and potential bias in the adjustment

One consideration when using this adjusted precipitation data set is the potential uncertainty and bias remaining in the observations following adjustment. Kochendorfer et al. (2018) pointed out that the uncertainty in post-adjusted precipitation data using WMO-SPICE transfer functions increased with the relative magnitude of the adjustment, showing that RMSE for a single Alter-shielded adjusted measurement nearly tripled (from ~0.2 to ~0.6 mm) as gauge height wind speed increased towards the threshold ($U_t$) of 7.2 m s$^{-1}$. This serves as justification for increased shielding (e.g. the double Alter-shield which is being installed throughout the ECCC network during modernization efforts) which increases the $C_E$ for a given wind speed thereby decreasing the magnitude of required adjustments and therefore

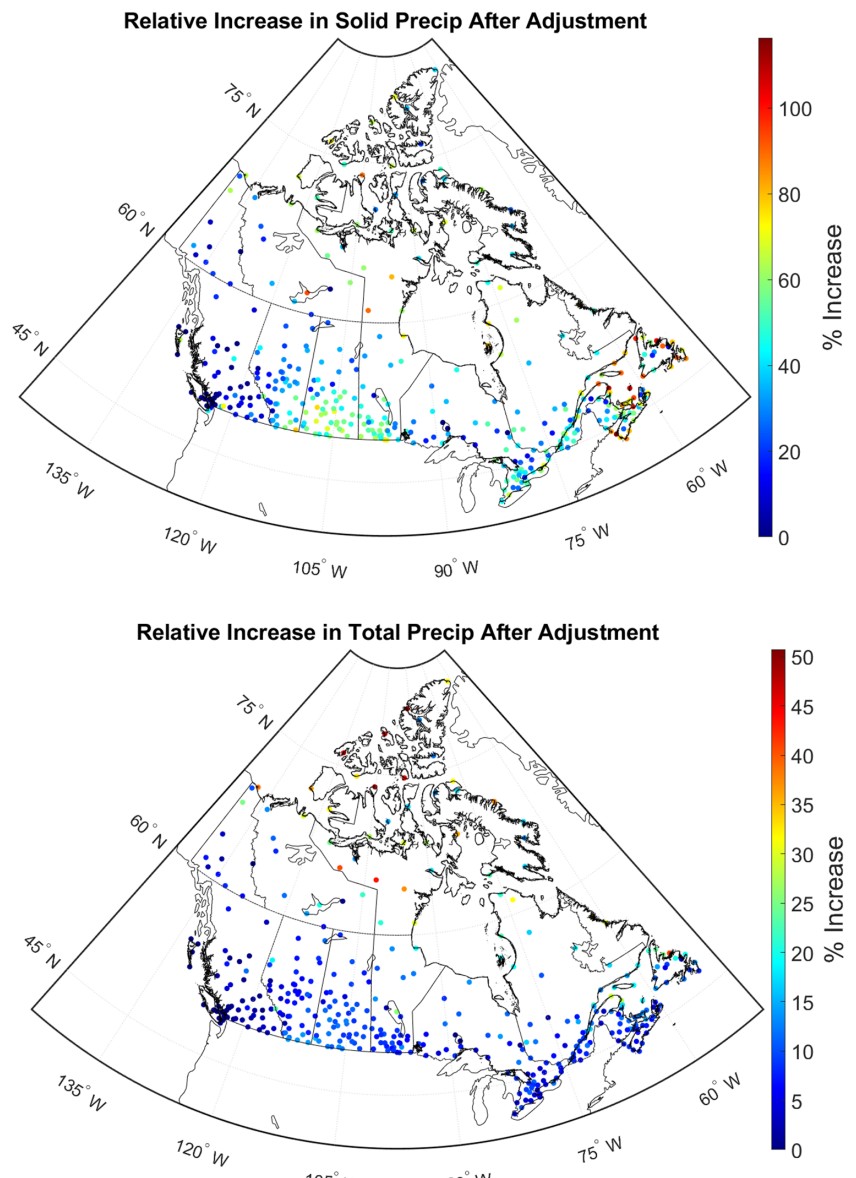

**Figure 6: Relative increase in station solid precipitation (top) and total precipitation (bottom) following adjustment for wind-bias using the WMO-SPICE UTF for the entire station record following automation.**



the uncertainty. Smith et al. (2020) showed that the post-adjusted RMSE (for solid precipitation) at Canadian test sites varied from 0.10 mm (low precipitation and exposed continental site) to 0.18 mm (higher precipitation and sheltered boreal site), very similar to the RMSE of the unadjusted observations (i.e. the adjustment did not increase the observation uncertainty).

Regarding the bias adjustment, it is obvious that a TF adjustment increases precipitation (especially solid precipitation) towards a "true" amount as represented by reference measurements. Smith et al. (2020) suggests that at cold and windy sites with low annual total and frequent light precipitation events, such as those in the Canadian Arctic and southern Prairies, the UTF tends to under-adjust solid precipitation by up to 30%. This would suggest that the adjusted winter precipitation in these regions remains low biased due to wind undercatch. On the other hand, the same analysis shows

that at wetter, less windy Canadian sites (based on test sites in southern boreal and wet continental climates), the UTF tends to over-adjust solid precipitation such that the adjusted amounts could be up to 110% of the reference amount.

### 3.4 Next steps

The application of the UTF on this set of 397 stations is the first step to achieve the overall objective of developing and applying climate dependent TFs on all stations across Canada. WMO-SPICE intercomparison data has been used

to fit site/climate specific transfer functions at eight SPICE sites (Køltzow et al., 2020; Kochendorfer, personal communication); work is in progress to develop more TFs further representing climates in Canada. However, no procedure currently exists to determine how to assign a climate specific TF to stations in such a climate diverse country as Canada. In order to sort and classify individual stations for adjustment, their climate characteristics must first be identified. These may be quantified with this adjusted precipitation data set and its accompanying wind and

temperature observations using two scenarios: 1) metrics calculated using the full station period, and 2) metrics calculated using data during precipitation occurrences. Also, options will be explored to match climate specific TFs to operational stations with qualitative climate characteristics from existing spatial climate classifications. Qualitative classification techniques may include but are not limited to the Köppen Climate Classification, the GIS-based Climatological Districts already used by ECCC, climate zones as defined in Climate Trends and Variation Bulletin,

and ecozones defined by the Canadian Council on Ecological Areas (CCEA).

It is anticipated that this adjusted precipitation data set will be updated periodically (the current data ends at the end of 2019) with new observations but also with the refinement of the TFs and the adjustment process. For this reason, this data set has been versioned as "v2019UTF".

**4 Applications**

This hourly wind-bias adjusted data set provides a wide range of opportunities in both climate and meteorological research areas.



Climate change detection needs long, reliable and homogeneous data sets. The AHCCD adjusted daily rainfall and snowfall data set (using the manual network) is available for 464 stations (Mekis et al, 2011) up to 2016; and the Adjusted Daily Rainfall and Snowfall (AdjDlyRS) data set (Wang et. al, 2017), which includes all 3346 segments from the manual archive, is available up to 2020. Built on the foundation of combining both manual and auto station observations, further development is underway to produce a more recent monthly adjusted and homogenized precipitation data set for 426 core stations. The hourly wind-bias adjusted data set will be linked into this data set for the overlapping locations.

The above mentioned AdjDlyRS 3346 adjusted daily stations were used to produce the ANUSPLIN gridded adjusted precipitation data set on daily, pentad and monthly time-scales (MacDonald et al, 2021). In the future, our new wind-bias adjusted automated observations will be used as an independent source for verification of the ANUSPLIN gridded precipitation data set.

Accurate and high-resolution analysis of precipitation accumulation is also primary importance for hydrological applications. In the last decade, the Canadian Precipitation Analysis (CaPA, Lespinas et. al, 2015; Asong et. al, 2017; Fortin et. al, 2018) system has been developed and implemented at MSC. It uses the optimal contributions of short-range forecasts from ECCC's Global Environmental Multiscale (GEM) numerical weather prediction (NWP) model and observations from surface stations and ground-based radars. There is, however, uncertainty with the input data used for solid precipitation, and with precipitation estimates in areas with few ground-based observations. The goal is to improve CaPa system performance by including adjusted solid precipitation, radar and satellite information to the system. The UTF hourly adjusted data set will be used for verification purposes in the above ongoing joint collaboration between Manitoba Hydro, University of Québec, Meteorological Research Division and Climate Research Division.

**5 Data availability**

Version v2019UTF of this data set, including documentation, is available via the Government of Canada Open Data Portal. To access the data, follow the DOI (https://doi.org/10.18164/6b90d130-4e73-422a-9374-07a2437d7e52; ECCC, 2021) to the metadata entry in the ECCC Data Catalogue. From the metadata page in the catalogue, the ECCC Data Mart links are shown by selecting the "Downloads, views, and links" option displayed below the data abstract. Following these links to the Data Mart will reveal the list of archived data and data description files available for download. The use of this data is subject to the Open Government License – Canada (https://open.canada.ca/en/open-government-licence-canada).





## 6 Summary

The hourly wind-bias adjusted precipitation data set from the ECCC automated surface observation network contains hourly wind-adjusted precipitation data as measured at 397 ECCC automated weather stations between 2001 and 2019. Based on WMO recommendations (Nitu et al., 2018), the hourly precipitation observations were adjusted using the

WMO-SPICE universal transfer function (UTF). Although we recognize that the SPICE UTF has performance issues that vary with site and climate (Smith et al., 2020), the high proportion of precipitation in Canada that falls as snow necessitates a measurement adjustment for undercatch due to wind. The adjusted precipitation data, including data documentation are available through the Government of Canada Open Data Portal (https://doi.org/10.18164/6b90d130-4e73-422a-9374-07a2437d7e52), currently versioned as v2019UTF (ECCC,

2021). This data set will be periodically updated with new data to include: 1) post-2019 adjusted precipitation, 2) additional stations in the ECCC automated network, and 3) additional adjusted segments using newly developed (double Alter-shield and climate specific) transfer functions.

## Author contribution

CS is the lead and corresponding author of this manuscript and provided the wind-bias adjustment. EM contributed to

the manuscript and provided expertise on the ECCC observation network and data quality control. MH provided quality control of the raw precipitation and met data and contributed to the manuscript. AR provided coding and data processing support and manuscript review.

## Competing Interests

The authors declare that they have no competing interest.

## Acknowledgements

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
