# Peer review of "The hourly wind-bias adjusted precipitation data set from the Environment and Climate Change Canada automated surface observation network (2001-2019)"

_Earth System Science Data, 2022_

## Referee Comment (RC1)

"The hourly wind-bias adjusted precipitation data set from the ECCC automated surface observation network" describes the evolution of ECCC precipitation measurements. It addresses the need to adjust some of the newer automated precipitation measurement for undercatch caused by wind. A new adjusted wind speed precipitation dataset is also described. The manuscript is well written and thorough. The need for wind adjustment is nicely described, in addition to the drawbacks of such adjustments.

Specific suggestions and questions:

Pg. 10, ln 15 – 23. It is surprising to me that unfiltered hourly total precipitation was not retained. Was this filter applied on site, before transmitting the data? It might be worth proving a little more detail on this, necessitating the regional hourly correction as the only method available to address this problem.

Pg. 8, ln 22 and 23. It is surprising to me that 10 m height winds are only available as the 2- or 10- min average prior to the top of the hour. As I am sure the authors are aware, it would be better to use an average wind speed that is representative of the entire hour over which precipitation occurred, and I suspect that nothing can be done about what is available. I am still curious as to why the wind speeds were recorded like this, and if nothing can be done to address this going forward. It might be worth evaluating the sensitivity of the adjustments to the choice of wind speed – there are plenty of other uncertainties in this type of adjustment, and in the end it may be impossible to say with certainty which approach is actually more accurate. However, it should be easy to quantify the uncertainty in Ce associated with using the less ideal 10 m wind; using the sites and periods when both wind speeds are available, a standard deviation (or whatever error statistic is preferred) could be calculated.

Pg 11, ln 5. "Ce is set to 1 at all temperatures greater than 5 deg C." Nicely done! The WMO-SPICE data does not include much warm/liquid precipitation. This is another good reason not to trust those adjustments for warm/liquid precipitation. Please point this out.

Pg 13, ln 11. Move the comma from after "but," to just before it. Like this: "…data set, but…"

Pg 14, ln 16. Change "effected" to, "affected."

Pg. 15, ln 17 – 18. I am gladdened and encouraged by the fact that at least one person understood the point of that figure I made for Kochendorfer et al. (2018)! However, the punctuation here is terrible. Add a comma after the closed parenthesis, "during moderation efforts),". And the "wind speed thereby" on ln. is awkward as written. At a minimum a comma is needed before "thereby", but it would be better to rephrase entirely. "Thereby… and therefore…". This can certainly be improved! Breaking this sentence into two might be a good start, or otherwise drastically changing the way it is structured.

Pg. 18, ln 12. Change, "also primary" to, "also of primary."

Pg. 18, ln 18. Add a comma after "radar." There are a other places where the Oxford comma is used inconsistently, but this one I find especially problematic.

---

## Author Response (AR1)

We would like to take this opportunity to thank Dr. John Kochendorfer and Anonymous Referee #2 for their time and effort to review our ESSD data description paper. The comments and suggested revisions are appreciated and will improve our manuscript. Author's responses to the referees comments are detailed below.

**Referee #1**

"The hourly wind-bias adjusted precipitation data set from the ECCC automated surface observation network" describes the evolution of ECCC precipitation measurements. It addresses the need to adjust some of the newer automated precipitation measurement for undercatch caused by wind. A new adjusted wind speed precipitation dataset is also described. The manuscript is well written and thorough. The need for wind adjustment is nicely described, in addition to the drawbacks of such adjustments.

Specific suggestions and questions:

**RC:** Pg. 10, ln 15 – 23. It is surprising to me that unfiltered hourly total precipitation was not retained. Was this filter applied on site, before transmitting the data? It might be worth proving a little more detail on this, necessitating the regional hourly correction as the only method available to address this problem.

**AC:** Unfortunately, this is the case and was a result of archaic but persistent data protocols for the ECCC operational network. The filter was applied between the data logger and the archive (but only on the hourly data and not on the 15-minute data when the 15-minute was available). The filtering error is described more by Milewska et al. (2019) but we will add more detail here.

**Action:** added the following sentences to the paragraph: "The filter was applied by the data management system, after the observations were logged by the data logger but before the data were archived. The pre-filtered logger data were not archived and therefore are not available."

**RC:** Pg. 8, ln 22 and 23. It is surprising to me that 10 m height winds are only available as the 2- or 10-min average prior to the top of the hour. As I am sure the authors are aware, it would be better to use an average wind speed that is representative of the entire hour over which precipitation occurred, and I suspect that nothing can be done about what is available. I am still curious as to why the wind speeds were recorded like this, and if nothing can be done to address this going forward. It might be worth evaluating the sensitivity of the adjustments to the choice of wind speed – there are plenty of other uncertainties in this type of adjustment, and in the end it may be impossible to say with certainty which approach is actually more accurate. However, it should be easy to quantify the uncertainty in Ce associated with using the less ideal 10 m wind; using the sites and periods when both wind speeds are available, a standard deviation (or whatever error statistic is preferred) could be calculated.

**AC:** We agree that it would be better to use a wind speed average that is representative of the entire period of the precipitation observation, and this is one reason why we flag which wind speed measurement is used in the adjustment. Fortunately, the 2 m wind speed measurements in the ECCC operational network are more representative of the full hour (and added to the network specifically to facilitate gauge wind-bias adjustments). We did some sensitivity analysis using wind speeds measured minutely at our non-operational research site at Bratt's Lake SK Canada. Using 1-minute wind speed data observed at 10 m over 3 continuous months starting in December 2015,  we calculated the hourly

wind speed using every 1-minute observation in the hour and compared it to the mean using only the mean from last 10 minutes in each hour. We then calculated a RMSE for the two techniques, which was determined to be about 0.5 m/s. This value was then used to calculate the sensitivity in the catch efficiency as it varies with wind speed and temperature (see Fig 2, right). This analysis showed that the biggest impact on CE was at low wind speeds and low temperatures (because this is when the slope of the CE relationship changes the fastest). At wind speeds close to 0 m/s at a temperature of -20° C, the difference in CE is less than 5%. At the site-average 10 m wind speed of 3.6 m/s, the difference in CE decreases to about 3%. Of course, this error may not be consistent from site to site, depending on the variability in wind speed with time. It is also difficult to put this uncertainty into context with the other large uncertainties related to the measurement and subsequent adjustment of precipitation.

**Action:** A note was made on Page 8 indicating that this item is discussed more in Section 3. Section 3.3 (page 15) was updated to include the above information about the uncertainty analysis related to the 10 m wind speed protocols.

**RC:** Pg 11, ln 5. "Ce is set to 1 at all temperatures greater than 5 deg C." Nicely done! The WMO-SPICE data does not include much warm/liquid precipitation. This is another good reason not to trust those adjustments for warm/liquid precipitation. Please point this out.

**AC:** Agreed.

**Action:** We added a sentence in Section 2.5 to point this out.

**RC:** Pg 13, ln 11. Move the comma from after "but," to just before it. Like this: "...data set, but..."

**AC:** Agreed

**Action:** this sentence was fixed

**RC:** Pg 14, ln 16. Change "effected" to, "affected."

**AC:** Agreed

**Action:** this was fixed

**RC:** Pg. 15, ln 17 – 18. I am gladdened and encouraged by the fact that at least one person understood the point of that figure I made for Kochendorfer et al. (2018)! However, the punctuation here is terrible. Add a comma after the closed parenthesis, "during moderation efforts),". And the "wind speed thereby" on ln. is awkward as written. At a minimum a comma is needed before "thereby", but it would be better to rephrase entirely. "Thereby... and therefore...". This can certainly be improved! Breaking this sentence into two might be a good start, or otherwise drastically changing the way it is structured.

**AC:** Agreed. Apologies for the poor punctuation and wording.

**Action:** these sentences were re-written to improve the readability.

**RC:** Pg. 18, ln 12. Change, "also primary" to, "also of primary."

**AC:** Agreed

**Action:** changed "also primary" to "also of primary."

**RC:** Pg. 18, ln 18. Add a comma after "radar." There are other places where the Oxford comma is used inconsistently, but this one I find especially problematic

**AC:** Agreed.

**Action:** the document was reviewed to check for consistent use of the Oxford comma and revisions made where required.

**Referee #2**

"The hourly wind-bias adjusted precipitation data set from the Environment and Climate Change Canada automated surface observation network (2001-2019)" provides a concise, comprehensive description of a new data set resulting from the evaluation and adjustment of hourly precipitation data at 397 automated climate stations across Canada. The authors provide a strong introduction, summarizing the issues with solid precipitation undercatch and justification for adjustments to precipitation measurements under various wind speeds and station configurations. Data and supporting documentation are available through a link provided in the manuscript. The documentation and station catalogue are comprehensive and informative, and the data set is high quality, complete, and useful. This manuscript was very well written and a pleasure to review. I have a few suggestions for revision, outlined below.

**RC:** Page 3, section 1.2. It would be beneficial to provide more background information on WMO-SPICE

**AC:** Although the WMO-SPICE project is well documented in Nitu et al. (2018), we agree that a few lines of project background would be informative for the reader.

**Action:** We have added some additional lines of background to the beginning of Section 1.2 before describing the recommendations, and added the link to the SPICE final report in the WMO library.

**RC:** Page 6, section 2.1. Additional stations were selected for proximity to AHCCD, as well as data quality and site longevity. What parameters or thresholds were used for data quality and site longevity?

**AC:** To clarify, the station selection was performed by our Environment and Climate Change Canada colleagues on a station-by-station basis based on proximity of new (post 2015) automated stations to earlier manual or AHCCD stations. ECCC scientists are using these stations in new data homogenization projects that are currently underway. Although the criteria for station selection is largely out of scope for this manuscript, our ECCC colleagues have stated that these will be published in more detail at a later time.

**Action:** We have revised Section 2.1 to read as follows:

"Initial station selection included all Geonor and Pluvio gauges that were installed prior to 2015 (Milewska et al., 2019). This included many stations in northern Canada, where automation started in the early 2000's. A later set of stations was added to this list, which were selected for their value to climate research. These later additions were selected from the automated stations installed post 2015, and included those that were in proximity to an earlier manual or adjusted and homogenized Canadian climate data (AHCCD, Mekis and Vincent, 2011) station. The stations selected are being used in  homogenization precipitation projects that are currently in progress. An effort was made to include all RCS locations, as they are designated as "protected" sites. "

**RC:** Page 14, line 9. Define TF; be consistent between using 'TF' or 'transfer function' throughout the manuscript

**AC:** Agreed.

**Action:** We have reviewed the manuscript to ensure consistency and changed "TF" to "transfer function" to refer to the more general catch efficiency curve but will continue to use "UTF" as an acronym for the more specific "universal transfer function".

[revised manuscript text omitted]
 potential source of uncertainty in the applied adjustment occurs when the data availability necessitate the use of 10 m rather than 2 m wind speeds. The concern is rooted in the methodology that MSC uses to determine hourly 10 m wind speed, which involves calculating the hourly mean wind speed using the last 10 minutes of the hour rather than the whole hour (see Section 2.3). To assess the sensitivity of $C_E$ to this potential error, three continuous months (starting Dec. 2015) of 1-min 10 m wind speed data from Bratt's Lake SK Canada (Smith, 2008) was used to calculate the hourly wind speed using 1) all of the 1-min data for each hour, and 2) the last 10 minutes of 1-min data for each hour. The RMSE for these two methods was 0.5 m s⁻¹ and resulted in a potential difference in $C_E$ of less than 5% at

low wind speeds (near 0 m s$^{-1}$ ) and at -20° C (when the rate of change in the transfer function curve is the highest, see Fig. 2, right). At the mean 10 m wind speed of 3.6 m s$^{-1}$, the potential error decreases to a difference in $C_E$ of 3%. Although not insubstantial, it is difficult to put this uncertainty into context with the other uncertainties involved in measuring and adjusting precipitation, especially considering that this potential error will vary with the site dependent variability in wind speed with time. The potential error, however, is minimized through the use of the 2 m wind speed (which is more representative of the whole hour) when available, and the adjustment is flagged accordingly.

AnotherOne consideration when using this adjusted precipitation data set is the potential uncertainty and bias remaining in the observations following adjustment. Kochendorfer et al. (2018) pointed out that the uncertainty in post-adjusted precipitation data using WMO-SPICE transfer functions increased with the relative magnitude of the adjustment, showing that RMSE for a single Alter-shielded adjusted measurement nearly tripled (from ~0.2 to ~0.6 mm) as gauge height wind speed increased towards the threshold ($U_t$) of 7.2 m s$^{-1}$. This is justification serves as justification for increased shielding around precipitation gauges, such as the (e.g. the double Alter-shield thatwhich 
[revised manuscript text omitted]

The authors would like to express their appreciation to Dr. John Kochendorfer and the anonymous referee for reviewing and providing feedback for the improvement of this manuscript. We would also like to thank Dr. Xiaolan Wang (Climate Research Division, ECCC) for providing the list of selected stations and the quality control protocols.

[revised manuscript text omitted]